# Adolescents’ Perceived Barriers to Physical Activity during the COVID-19 Pandemic

**DOI:** 10.3390/children9111726

**Published:** 2022-11-10

**Authors:** Carlos Mata, Marcos Onofre, João Martins

**Affiliations:** 1Centro de Estudos de Educação, Faculdade de Motricidade Humana, Universidade de Lisboa, 1499-002 Cruz Quebrada-Dafundo, Portugal; 2Centro de Estudos de Educação, Faculdade de Motricidade Humana e UIDEF, Instituto de Educação, Universidade de Lisboa, 1649-004 Lisboa, Portugal; 3CIPER, Faculdade de Motricidade Humana, Universidade de Lisboa, 1499-002 Cruz Quebrada-Dafundo, Portugal

**Keywords:** physical activity, barriers, adolescents, COVID-19

## Abstract

During the COVID-19 pandemic, adolescents’ routines were deeply affected, which negatively impacted their level of PA. Knowing the barriers to PA in adolescence is relevant, because the perception of more barriers is one of the most consistent negative correlates of PA participation. The purpose of this study was to analyze and compare the barriers perceived by adolescents during the COVID-19 pandemic by sex, education level, PA level, and BMI. A total of 1369 students (621 boys and 748 girls; mean age: 14.4 years; SD: 1.74) participated in the study. The chi-square test was used to analyze the differences between groups. Only 3.1% of the adolescents complied with the international guidelines for PA. In general, the barriers with the highest prevalence were the COVID-19 pandemic, lack of time, and taking time away from study. The number of perceived barriers to PA was higher among girls, younger, and inactive participants. Boys selected more the barriers *due to COVID-19* than girls, and students with normal weight chose more barriers than those with overweight. This study provides information on adolescents’ PA barriers during the COVID-19 pandemic and draws attention to the negative effects that restrictive measures have had on adolescents’ PA levels.

## 1. Introduction

There is evidence that regular physical activity (PA) reduces the risk of premature mortality and is an effective primary and secondary preventive strategy for a range of chronic medical conditions [1,2]. In adolescents, regular engagement in PA provides physical, psychological, social, and cognitive health benefits [3,4]. To achieve these benefits, it is recommended that adolescents engage in 60 min of daily moderate to vigorous PA [5]. However, despite the wide dissemination of the importance of adopting an active and healthy lifestyle, most children and adolescents do not meet the guidelines of PA [6,7]. Furthermore, participation in regular PA decreases with age, with this being a problem more noticeable in girls, who are less active than boys in almost all countries worldwide (84.7%, compared to 77.6% for boys), a trend that has been stable over the last decade [6]. Moreover, this global physical inactivity trend at these ages is particularly relevant and of concern, because PA habits adopted at a young age will tend to remain throughout life [8].

In March 2022, when COVID-19 was considered a pandemic [9], several restrictive measures were imposed to combat the spread of the virus, leading thousands of children and adolescents to stop attending school, which significantly changed their daily routines. This complete lockdown and the impossibility of any social contact outside the family context, resulted in a decrease in PA participation levels and an increase in sedentary behaviors and screen time among young people [10,11,12,13,14].

Several studies have pointed out impacts in different health domains as a result of the pandemic and the prolonged lockdown, namely, physical, mental, emotional, and behavioral [15,16,17,18]. In Portugal, in September 2022, students returned to school and had to comply with a set of safety rules, which significantly impacted their school and extra school life [19]. During this period, many clubs and sports associations closed, limiting the possibilities of PA practice to the school context. In most cases, PE classes became the only possible context for thousands of students to participate in formal PA, even though strongly conditioned by the rules adopted to reduce contagion [20]. This set of negative factors may have had an effect of increasing exposure to risks related to insufficient levels of PA and the adoption of sedentary behaviors. In addition, it should be noted that PA has multiple, well-documented benefits related to reducing the impact of the severe acute respiratory syndrome coronavirus 2 (SARS-CoV-2) infection itself, as well as helping to fight social isolation and stress caused by the pandemic [21]. This fact should be enough for PA-promoting policies to be taken during this critical period, which generally was not the case, since most governments have opted for restrictive measures and there have been no incentive measures to mitigate the negative impacts of the increase in inactivity.

Most studies looking at the impact of COVID-19 on adolescents’ PA levels related to periods of lockdown, which varied from country to country according to the measures taken and pandemic expression in each situation [18]. The evidence, in most studies, even if conducted at different pandemic moments, shows an overall decrease in engagement in PA [18,22,23].

Several studies identify factors that influence adherence to PA during adolescence, including demographic variables, attitudes, knowledge, and beliefs about PA [24]. The reasons why individuals do not engage in PA are complex and multifactorial, involving personal and environmental determinants and other external factors [24,25]. The social-ecological model [26] emerges as an appropriate method when aiming to understand the complexity of the interaction of individual, physical, social, and environmental factors of PA behavior. We should highlight, within the personal factors, overweight/obesity as one of the markedly important factors in adherence to PA. The high prevalence of overweight in children and adolescents has become a global trend and is one of the most serious public health problems of the 21st century [27]. In addition to health problems that are associated with youth obesity and the future risks it represents, overweight children and adolescents tend to engage less in PA [28,29], “wasting” the positive effect that regular PA practice can have in reducing childhood obesity [30,31,32,33]. Furthermore, for overweight and obese children and adolescents, PA barriers may be perceived as greater or even more challenging, which may result in a more pronounced decrease in PA engagement level [34]. Perceiving more barriers to PA is, in fact, one of the most consistent negative correlates of children’s PA participation [35,36].

The barriers most frequently identified (in quantitative and qualitative studies) by young people to being physically active are: negative experiences of PA at school and PE; personal factors (e.g., motivation, self-consciousness about appearance); constraints related to family and friends; lack of time; high cost of equipment and low accessibility or availability of facilities [25,37]. In adolescence, there are differences in the perception of PA barriers, between boys and girls, between younger and older, and among young people from different socioeconomic backgrounds, facts that should be considered in the adoption of PA-promotion strategies in this population [38], including in times of pandemics.

We may assume that during the pandemic, the identified barriers to participation in PA might be different or have distinctive relevance when compared to the pre-pandemic time, considering that adolescents’ routines were significantly changed. The perceptions may also vary according to the moment when the data are collected, as daily routines during the lockdown were certainly different from those experienced during the pandemic after returning to school.

In a recent review of children and adolescents’ PA during the COVID-19 pandemic, the barriers to PA participation are explained on two levels: individual and context-specific [18]. At both levels, we found factors more specific to the pandemic moment, such as: feeling comfortable at home, pre-COVID sedentary time, pre-COVID activity levels or feelings of stress (individual); restrictions from COVID-19, club training cancellation, or lack of playmates (context) [18].

Data on the effect of restrictions and limitations in Portugal, after confinement and during the pandemic, on the PA level of young people are few or nonexistent, to the best of our knowledge. In our study, data were collected after the return to school, but still in the middle of the pandemic, with many restrictive measures being implemented to avoid contagion and with students experiencing many changes in their school routines. It is important to know the barriers of PA in this critical moment, in order to design more efficient PA interventions in future pandemics that may occur, and to help children and adolescents, especially girls, to be more active. It is also relevant to understand if there are differences between groups at this moment (e.g., boys and girls, younger and older) and thus adapt the strategies of PA professionals to these groups, helping to overcome the specific barriers revealed by adolescents. Therefore, this study aims to understand what barriers to PA adolescents identify in this critical and unprecedented period, comparing the perceptions of boys and girls, more and less active, according to different school levels of education (elementary vs. secondary) and according to students’ BMI.

## 2. Materials and Methods

### 2.1. Participants

Data were gathered from students aged 12–18 years (Mean: 14.4 years; SD: 1.74) from 13 schools in the north of Portugal, belonging to levels two (lower secondary) and three (upper secondary) of the International Standard Classification of Education (ISCED). The schools belonged to different urban and socioeconomic contexts and were chosen by convenience. Only students who were attending PE classes at the time the survey was applied participated in the study.

### 2.2. Procedures

The LimeSurvey platform was used to complete the online survey that took place between November and December 2020, when students returned to face-to-face teaching, after the school closures in March 2020.

All the operational and ethical procedures for data collection were followed: contacts with PE teachers who collaborated in the study (October 2020), in which the objectives and application process were explained; information was provided to the school principals and to the guardians of the participating students. Only students duly authorized by their guardians participated in the study and all issues regarding confidentiality and data protection were ensured. Students were informed that their participation was voluntary, and they could interrupt or stop it at any time. The link with the questionnaire was sent by e-mail to the PE teachers and the electronic completion was performed in a PE class, with the presence of their teacher or another PE teacher from the school.

Pilot tests were previously conducted with a group of 15 high school students, in order to assess difficulties in understanding the questions and to check how long it took them to answer the questionnaire. The mean response time was about 25 min.

This survey was approved by the Ethics Council for Research of the host institution (Log No. 16/2020) and by the Ministry of Education of Portugal Government (Log No. 0666900005, approved on March 2020).

In the PA assessment, we adjusted the second item by adding the phrase “before COVID-19” in order to understand the pre-pandemic PA patterns. In line with the PA frequency assessment of Prochaska et al. [39], the answers to the two questions regarding weekly PA frequency ranged from “0 to 7 days”. We added an introductory note (before the two questions) alluding to the aspects that students should consider when assessing their PA level—“Physical activity is any activity that increases your heart rate and makes you feel breathless. Physical activity can be done playing sports, in school activities, playing with friends, or walking. Some examples of physical activity are running, biking, dancing, skateboarding, swimming, playing basketball, playing soccer, and surfing”. Question 1: “In the past 7 days, on how many days did you engage in physical activity for a total of at least 60 min per day?”; Question 2: “In a normal week before COVID-19, on how many days did you engage in at least 60 min of physical activity per day?” Only students who met the WHO guidelines were considered “active”.

Weight, height, age, sex, and year of schooling were assessed by self-report. BMI was calculated (BMI: kg/m^2^) and the BMI z-score was adjusted for age and sex, according to the WHO reference values [40].

For the assessment of PA barriers, we chose a set of 22 barriers, 21 already used in other quantitative studies plus the *due to COVID-19 pandemic* barrier. These barriers have already been used in studies with adolescents in Portugal [41,42]. Participants were informed that there was no limit on the number of selections from the set of 22 barriers.

The results were extracted from the LimeSurvey platform into Excel, creating the database that was exported to SPSS, version 27 for MacOS (SPSS Inc., Chicago, IL, USA), where the statistical analysis was performed. In the descriptive analyses, absolute (n) and relative (%) frequencies were calculated for categorical variables and means (M), and standard deviations (SD) for continuous variables. The association of categorical variables, namely barriers to PA with sex, education level, active/inactive students, and standardized BMI was assessed with the chi-square test. In the case of non-compliance with the test requirements, Fisher’s test was applied. The significance level was set at 0.05.

## 3. Results

The sample characteristics are presented in Table 1, with values expressed as mean and standard deviation or *n* cases (%) of: participants (and the corresponding education level to which they belong); participants’ age; meeting or not the guidelines for PA before and during the COVID-19 pandemic. We underline the low percentage of participants who complied with the 60-min daily PA guidelines before the pandemic (4.9%). During the pandemic, the value was even lower (3.1%). Compared to boys, girls presented lower PA levels.

Barriers to PA were stratified by sex, education level, meeting PA guidelines before COVID-19, and standardized BMI. Overall, the barriers with the highest prevalence were lack of time, taking time away from study, and the COVID-19 pandemic.

Table 2 shows that girls significantly selected a higher percentage of barriers compared to boys: *lack of time* (*p* < 0.001), *lack of ability* (*p* < 0.001), *I’m ashamed*, *because it takes away time that I need to study* (*p* < 0.001), *I don’t like competition* (*p* < 0.001), *I don’t have an accessible place or club to do PA where I live* (*p* = 0.022), *lack of motivation/interest* (*p* < 0.001), and *I’m afraid* (*p* = 0.047). Boys significantly selected a higher percentage of the *due to COVID-19 pandemic* barrier (*p* < 0.001) than girls.

Table 3 shows the selected barriers to PA, by level of education. Younger students (ISCED 2) showed a significantly higher proportion in barriers related to *lack of time* (*p* < 0.001), *don’t like to sweat* (*p* = 0.007), *because it takes away time that I need to study* (*p* = 0.003), *lack of motivation/interest* (*p* = 0.042), *because it takes away time to spend with friends* (*p* = 0.041), and *I’m afraid of getting injured* (*p* = 0.009).

Table 4 compares the groups of inactive and active students. The inactive group selected a higher proportion of PA barriers, namely, *lack of time* (*p* = 0.042), *I’m not in shape* (*p* = 0.010), *there are more interesting things to do* (*p* = 0.023), and *lack of motivation/interest* (*p* = 0.005).

To evaluate the association of barriers with BMI, two groups were considered: normal weight (85.5%) and overweight (15.5%).

Table 5 shows that the normal weight group selected significantly more PA barriers than the overweight subjects, namely, *lack of time* (*p* = 0.047), *I’m not in shape* (*p* < 0.001), *lack of ability* (*p* < 0.001), *I’m ashamed* (*p* < 0.001), *too expensive* (*p* = 0.017), *I don’t like competition* (*p* = 0.005), and *lack of motivation/interest* (*p* = 0.002).

## 4. Discussion

The results showed a low percentage of participants meeting the PA guidelines, [43], which is below the results presented in recent studies with Portuguese adolescents [44,45] and also from the available international data [6]. The percentage of adolescents who met the PA guidelines during the pandemic dropped to 3.1%, in agreement with most studies that assessed PA level during the pandemic, where a decrease in PA engagement was reported [18,22]. In both situations, girls had a lower percentage of PA participation than boys, which is in line with the general trend of girls’ lower involvement in PA [6,44,46]. The high prevalence of physical inactivity, its detrimental health and environmental effects, and the evidence of positive outcomes of PA-promoting strategies, make this problem an urgent and priority public health issue [47]. On the other hand, it highlights the special attention that should be paid during pandemic periods, since, as mentioned, the level of participation in PA reduces even more. The emerging evidence from studies conducted during the pandemic, show a trend of decreasing levels of PA in children and adolescents and increasing sedentary behaviors, with potential negative consequences in different health domains [17,18,22].

The number of perceived barriers to PA were higher among girls than boys, as it is reported in many studies [25,38,48]. Girls significantly reported *lack of time* and *lack of time to study* as the most relevant barriers to not engaging in PA in this period, as has been mentioned in the literature [25,49,50,51]. *Lack of motivation/interest* and *lack of ability* also appear as relevant and significant barriers perceived by girls at this particular time. Reviews also report these barriers perceived by girls as significant [25,49,50], which seems to indicate that in times of pandemics these perceptions have remained stable. Girls also chose significantly more individual barriers than boys, namely: *I don’t like competition*, *I’m ashamed*, and *I’m afraid*. The interpersonal barriers pointed out by girls in different studies seem to have no special relevance in this pandemic context. For example, the lack of peer and family support is pointed out in different studies as a barrier with impacts on adherence to PA [25,49,50]. This may be explained by the change in routines and reduced social contact, which may have conditioned the girls’ perception of the obstacles to PA, valuing individual barriers more than interpersonal ones. It is, however, relevant to mention that the third most selected barrier by girls was *due to COVID 19 pandemic*. The most well-known barriers and frequently pointed out in studies appeared again in this pandemic period, but a new barrier emerges, which is precisely linked to COVID-19.

Within PA environmental barriers, the lack of facilities to practice PA emerges in the literature as an obstacle to girls’ participation [25,51]. During the pandemic, in Portugal, the possibilities for children and adolescents to engage in PA were significantly reduced, due to the closure of sports clubs and the prohibition of access to recreational areas, which may have induced the significantly higher girls’ choice for the barrier *I don’t have an accessible place or club to do PA where I live*. This may not be directly related to the pandemic, as this perception could have existed before the restrictions, but draws attention to the importance of promoting safe and favorable conditions for access to facilities for PA, formal or recreational.

Although girls also strongly selected the barrier *due to COVID-19 pandemic*, boys were probably more affected by the restrictions applied during this period, as the percentage of boys who selected this barrier was significantly higher than girls. This might reveal a greater impact of the pandemic on boys and that they may have felt a greater negative effect on their PA routines, as they experienced limited access to sports clubs, suppression of extracurricular activities at school (e.g., school sports), and also by being unable to perform informal PA in and out of school, due to the social distance. This highlights the importance that must be given to policies that lead to increased opportunities for adolescents to engage in PA in times of a pandemic and these strategies should take into consideration the gender.

Comparing perceived barriers to PA between younger and older students (ISCED 2 and ISCED 3), we found that younger participants selected significantly more barriers to PA. Most of the barriers chosen by the younger students were related to *lack of time* (time in general, to study and to be with friends) and *lack of motivation*. In this comparison between older and younger participants, individual PA barriers are again the most prominent, as we still find *don’t like to sweat* and *I’m afraid of getting injured* as significantly higher in the younger than in the older ones. Apparently, the younger students felt most acutely all the restrictions and prohibitions they had to go through during this difficult period. In a different study, but with the same sample, lower levels of motivation and perceived motivational climate were identified in younger students in PE classes during the pandemic [14]. The results found in our study confirm this to some extent. It may also add more data about the impact of the COVID-19 pandemic on adolescents, particularly in the younger ones, which seemed to be more profound.

When perceptions of PA barriers were compared between inactive and active students, we found that inactive students selected significantly more barriers than the active ones, focusing their choices on individual barriers: *lack of time*, *I’m not in shape*, *there are more interesting things to do*, and *lack of motivation/interest*. Although not significant, when compared to the active ones, almost half of the inactive (49.6%) and a high percentage of the active (61.2%) students selected COVID-19 as the main PA barrier during this period. The study by Ng et al. [23], while referring to barriers to PA during the lockdown in Ireland due to COVID-19, identified a greater impact of the pandemic on less active adolescents. The COVID-19 restrictions as well as the cancellation of training sessions in clubs were the most common barriers mentioned in this study by adolescents. In our study, we have to point out that students had returned to school when we collected the data, and that this may have changed their perception of what they considered to be barriers to PA, compared to when they were in lockdown.

It is suggested in the literature that overweight adolescents perceive more barriers than those of normal weight, regardless of the barrier type [28,34]. In this study, although the differences are not significant, about half of the normal weight (49.8%) and overweight (51.4%) students selected COVID-19 as the most important barrier for PA involvement. When comparing normal weight and overweight students, we highlight another relevant aspect: participants with normal weight selected significantly more PA barriers than those with overweight. Apparently contradictory to the literature, this might be related to the specific period in which the study was carried out, with students affected by the profound changes in their school life and daily routines. It is likely that normal weight adolescents perceived more barriers to PA because they felt their structures were affected during this period. The PA performed in Portugal by children and adolescents was essentially limited to PE classes, which also operated with strong limitations in their organization [20]. The constant need to maintain social distance led to the suppression of team sports, focusing class activities on individual sports and physical fitness, probably generating less motivation in students, independently of their weight being within normal standards or not, blurring differences in PA engagement level between normal weight and overweight students.

The barrier *lack of time* was significantly identified by both groups. Although with higher incidence in the normal weight group, the percentage of overweight adolescents who chose this barrier was equally high. Although the barrier *lack of time* is mentioned in many studies as one of the barriers most often selected by adolescents [25,38,52], in the context of the pandemic it may assume a different dimension. In this study, *lack of time*, is also frequently mentioned as an important barrier, and this might be associated with the extra pressure students felt when they returned to school after the lockdown. One of the measures adopted by schools in Portugal during the COVID-19 pandemic, following the government’s determinations, was to recover study subjects from the curriculum that were left unaccomplished during the online classes, as not all students had the same opportunities to follow the classes during the lockdown (lack of computers, poor internet connection, difficulties in adapting to a new reality, etc.). This extra work may have led to a greater perception that time for activities other than academic ones was scarce. Aside from this argument, it is important to reflect on the fact that the barrier lack of time is one of the most mentioned in studies. As argued by Biddle [53], lack of time may be a perception rather than a reality. When asked about barriers to PA, lack of time will emerge as one of the most relevant, if subjects have already allocated their free time to alternative and more reinforcing behaviors. This reveals, fundamentally, a hierarchy of values in which PA is far down the list to be allocated time devoted to it [53].

As mentioned by Duffey et al. [38], although the barriers can be analyzed separately, it is important to note the interaction and complexity of each and how they affect adolescents’ PA participation when aiming to intervene to improve participation levels. This may provide better guidance in designing multi-component interventions, targeting multiple influencing factors that address these complex factors affecting adolescents’ PA participation.

The severe limitations in school life experienced by adolescents in the midst of the pandemic, reveal data that contradict the literature in part (before COVID-19). Whole new experiences may have led to distinct perceptions regarding PA barriers. Strategies to promote PA should be tailored to the elements that are identified as potential barriers to behavior change. The barriers, both external (e.g., environmental factors, availability of sports facilities, social supports) and internal (e.g., motivation and interests), can indeed influence the efficiency of preventive intervention programs [48] and this is true for a normal context or for pandemic periods.

This study presents some limitations, which we describe: PA was assessed by self-report, which may lead to results that are not as accurate compared to assessment by objective methods, particularly among young people. In any case, we are not aware of any studies in Portugal that have objectively assessed PA levels during the pandemic, after the first lockdown, which does not allow us, so far, to compare data with other PA measures. Data from this study cannot be extrapolated, as they reflect a unique and never experienced context. Evidence points to girls’ lower involvement in PE. The fact that the questionnaire was applied in PE classes may have in itself skewed the results for girls to illustrate low participation.

This study has, however, strengths that should be mentioned: sample size, data were collected when students returned to school, in an historically unique period, and PA studies with adolescents during the pandemic after the first lockdown are rare or non-existent. Considering that the existing literature depicts different moments of the pandemic and that different phases correspond to distinct measures to fight the virus, future studies should focus on the outcomes of decreased physical activity to create a more holistic picture of the effects of the COVID-19 pandemic on children and adolescents.

## 5. Conclusions

The results of the study support that only 3.1% of the adolescents complied with the PA guidelines. The perceived barriers with the highest prevalence were the COVID-19 pandemic and lack of time. Differences were found based on sex, level of education, PA levels, and weight status. It is important to identify the barriers to PA perceived by adolescents during critical moments, such as pandemics, because situations of restriction and lockdown may occur again. Policies and PA-promoting measures should be adopted to mitigate the limitations and privations that adolescents face during pandemic events or other life-altering disaster situations, with implications on the level of PA and with impacts on several health domains.

## Figures and Tables

**Table 1 children-09-01726-t001:** Sample description stratified by sex.

	ISCED 2		ISCED 3		Total		Age		^1^ PA		^2^ PA	
	n	%	n	%	n	%	M	SD	n	%	n	%
Male	385	62.0	236	38.0	621	45.4	14.3	1.74	43	6.9	26	4.1
Female	432	57.8	316	42.2	748	54.6	14.4	1.74	24	3.2	17	2.2
Total	817	59.7	552	40.3	1369	100	14.4	1.74	67	4.9	43	3.1

^1^ PA—performs 60 min of PA every day before COVID-19; ^2^ PA—does not perform 60 min of PA every day before COVID-19. ISCED 2—lower secondary education; ISCED 3—upper secondary education; M—mean; SD—standard deviation.

**Table 2 children-09-01726-t002:** PA barriers stratified by sex.

Barrier	Boys: n (%)	Girls: n (%)	Χ^2^ Test
Due to COVID-19 pandemic	372 (59.9%)	315 (42.1%)	** *p* ** **< 0.001**
Lack of time	226 (36.4%)	430 (57.5%)	** *p* ** **< 0.001**
Because it takes away time that I need to study	165 (26.6%)	383 (51.2%)	** *p* ** **< 0.001**
Lack of motivation/interest	88 (14.2%)	176 (23.5%)	** *p* ** **< 0.001**
Lack of ability	52 (8.4%)	116 (15.5%)	** *p* ** **< 0.001**
I’m not in shape	86 (13.8%)	98 (13.1%)	*p* = 0.687
There are more interesting things to do	93 (15.0%)	96 (12.8%)	*p* = 0.253
I don’t have an accessible place or club to do PA where I live	54 (8.7%)	94 (12.6%)	** *p* ** **= 0.022**
I don’t like competition	35 (5.6%)	88 (11.8%)	** *p* ** **< 0.001**
I’m afraid of getting injured	44 (7.1%)	73 (9.8%)	*p* = 0.078
I don’t have friends to do PA with	46 (7.4%)	71 (9.5%)	*p* = 0.170
Because it takes away time to spend with friends	69 (11.1%)	65 (8.7%)	*p* = 0.133
I’m ashamed	24 (3.9%)	65 (8.7%)	** *p* ** **< 0.001**
Don’t like to sweat	45 (7.2%)	56 (7.5%)	*p* = 0.866
I’ve got physical limitations	43 (6.9%)	54 (7.2%)	*p* = 0.832
The others are better	31 (5.0%)	53 (7.1%)	*p* = 0.108
I need equipment that I do not have	37 (6.0%)	39 (5.2%)	*p* = 0.549
Too expensive	24 (3.9%)	35 (4.7%)	*p* = 0.460
PA opportunities are not interesting	28 (4.5%)	26 (3.5%)	*p* = 0.328
I’m afraid	9 (1.4%)	23 (3.1%)	** *p* ** **= 0.047**
My parents don’t allow	14 (2.3%)	12 (1.6%)	*p* = 0.380
Doing PA/sport is a boy thing and is not for girls	9 (1.4%)	4 (0.5%)	*p* = 0.082

Values in brackets correspond to the percentage within sex. Results with statistical significance are presented in bold.

**Table 3 children-09-01726-t003:** PA barriers stratified by level of education.

Barrier	ISCED 2	ISCED 3	Χ^2^ Test
Due to COVID-19 pandemic	423 (51.8%)	264 (47.8%)	*p* = 0.152
Lack of time	350 (42.8%)	306 (55.4%)	** *p* ** **< 0.001**
Because it takes away time that I need to study	301 (36.8%)	247 (44.7%)	** *p* ** **= 0.003**
Lack of motivation/interest	143 (17.5%)	121 (21.9%)	** *p* ** **= 0.042**
I’m not in shape	105 (12.9%)	79 (14.3%)	*p* = 0.437
There are more interesting things to do	117 (14.3%)	72 (13.0%)	*p* = 0.502
Lack of ability	106 (13.0%)	62 (11.2%)	*p* = 0.335
I don’t have an accessible place or club to do PA where I live	92 (11.3%)	56 (10.1%)	*p* = 0.514
Because it takes away time to spend with friends	91 (11.1%)	43 (7.8%)	** *p* ** **= 0.041**
I’m afraid of getting injured	83 (10.2%)	34 (6.2%)	** *p* ** **= 0.009**
I don’t like competition	83 (10.2%)	40 (7.2%)	*p* = 0.064
Don’t like to sweat	73 (8.9%)	28 (5.1%)	** *p* ** **= 0.007**
I don’t have friends to do PA with	63 (7.7%)	54 (9.8%)	*p* = 0.179
I’ve got physical limitations	55 (6.7%)	42 (7.6%)	*p* = 0.535
The others are better	53 (6.5%)	31 (5.6%)	*p* = 0.510
I’m ashamed	50 (6.1%)	39 (7.1%)	*p* = 0.487
I need equipment that I do not have	41 (5.0%)	35 (6.3%)	*p* = 0.295
PA opportunities are not interesting	32 (3.9%)	22 (4.0%)	*p* = 0.949
Too expensive	28 (3.4%)	31 (5.6%)	*p* = 0.050
I’m afraid	21 (2.6%)	11 (2.0%)	*p* = 0.488
My parents don’t allow	16 (2.0%)	10 (1.8%)	*p* = 0.845
Doing PA/sport is a boy thing and is not for girls	8 (1.0%)	5 (0.9%)	*p* = 0.891

ISCED 2—lower secondary education; ISCED 3—upper secondary education. Values in brackets correspond to the percentage within each level of education. Results with statistical significance are presented in bold.

**Table 4 children-09-01726-t004:** PA barriers stratified by inactive and active students.

Barrier	Inactive	Active	Χ^2^ Test
Due to COVID-19 pandemic	646 (49.6%)	41 (61.2%)	*p* = 0.065
Lack of time	632 (48.5%)	24 (35.8%)	** *p* ** **= 0.042**
Because it takes away time that I need to study	527 (40.5%)	21 (31.3%)	*p* = 0.137
Lack of motivation/interest	260 (20.0%)	4 (6.0%)	** *p* ** **= 0.005**
There are more interesting things to do	186 (14.3%)	3 (4.5%)	** *p* ** **= 0.023**
I’m not in shape	182 (14.0%)	2 (3.0%)	** *p* ** **= 0.010**
Lack of ability	163 (12.5%)	5 (7.5%)	*p* = 0.219
I don’t have an accessible place or club to do PA where I live	143 (11.0%)	5 (7.5%)	*p* = 0.365
Because it takes away time to spend with friends	128 (9.8%)	6 (9.0%)	*p* = 0.814
I don’t like competition	118 (9.1%)	5 (7.5%)	*p* = 0.655
I don’t have friends to do PA with	115 (8.8%)	2 (3.0%)	*p* = 0.095
I’m afraid of getting injured	111 (8.5%)	6 (9.0%)	*p* = 0.902
Don’t like to sweat	96 (7.4%)	5 (7.5%)	*p* = 0.978
I’ve got physical limitations	90 (6.9%)	7 (10.4%)	*p* = 0.271
I’m ashamed	85 (6.5%)	4 (6.0%)	*p* = 0.857
The others are better	81 (6.2%)	3 (4.5%)	*p* = 0.562
I need equipment that I do not have	70 (5.4%)	6 (9.0%)	*p* = 0.212
Too expensive	55 (4.2%)	4 (6.0%)	*p* = 0.493
PA opportunities are not interesting	52 (4.0%)	2 (3.0%)	*p* = 0.679
I’m afraid	30 (2.3%)	2 (3.0%)	*p* = 0.719
My parents don’t allow	24 (1.8%)	2 (3.0%)	*p* = 0.504
Doing PA/sport is a boy thing and is not for girls	11 (0.8%)	2 (3.0%)	*p* = 0.078

Inactive—does not perform 60 min of PA every day before COVID 19; Active—performs 60 min of PA every day before COVID-19. Values in brackets correspond to the percentage within the inactive and active groups. Results with statistical significance are presented in bold.

**Table 5 children-09-01726-t005:** PA barriers stratified by BMI (two categories).

Barrier	Normal	Overweight	Χ^2^ Test
Due to COVID-19 pandemic	583 (49.8%)	95 (51.4%)	*p* = 0.660
Lack of time	573 (48.9%)	76 (41.1%)	** *p* ** **= 0.047**
Because it takes away time that I need to study	475 (40.6%)	70 (37.8%)	*p* = 0.292
Lack of motivation/interest	222 (19.0%)	40 (21.6%)	** *p* ** **= 0.002**
There are more interesting things to do	162 (13.8%)	25 (13.5%)	*p* = 0.482
Lack of ability	121 (10.3%)	41 (22.2%)	** *p* ** **< 0.001**
I’m not in shape	120 (10.2%)	61 (33.0%)	** *p* ** **< 0.001**
I don’t have an accessible place or club to do PA where I live	116 (9.9%)	31 (16.8%)	*p* = 0.143
Because it takes away time to spend with friends	116 (9.9%)	13 (7.0%)	*p* = 0.905
I don’t like competition	99 (8.5%)	20 (10.8%)	** *p* ** **= 0.005**
I’m afraid of getting injured	98 (8.4%)	15 (8.1%)	*p* = 0.150
I don’t have friends to do PA with	95 (8.1%)	21 (11.4%)	*p* = 0.394
Don’t like to sweat	87 (7.4%)	10 (5.4%)	*p* = 0.321
I’ve got physical limitations	76 (6.5%)	19 (10.3%)	*p* = 0.061
The others are better	66 (5.6%)	15 (8.1%)	*p* = 0.187
I’m ashamed	62 (5.3%)	24 (13.0%)	** *p* ** **< 0.001**
I need equipment that I do not have	55 (4.7%)	19 (10.3%)	*p* = 0.798
PA opportunities are not interesting	46 (3.9%)	8 (4.3%)	*p* = 0.215
Too expensive	44 (3.8%)	14 (7.6%)	** *p* ** **= 0.017**
I’m afraid	23 (2.0%)	6 (3.2%)	*p* = 0.692
My parents don’t allow	22 (1.9%)	3 (1.6%)	*p* = 0.809
Doing PA/sport is a boy thing and is not for girls	13 (1.1%)	0 (0.0%)	*p* = 0.271 *

* Fisher’s exact test. Values in brackets correspond to the percentage within the normal and overweight groups. Results with statistical significance are presented in bold.

## Data Availability

The data presented in this study are available on request from the corresponding author. The data are not publicly available as they were collected, processed, and calculated by the author.

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
