# Peer review of "Adolescents’ Perceived Barriers to Physical Activity during the COVID-19 Pandemic"

_children, 2022, doi:10.3390/children9111726_

Round 1

Reviewer 1 Report

This is an interesting topic area of interest to the reader. A few amendments required and attention to the tables in the results section is a necessity.

The title needs to read: ‘Adolescents’ perceived barriers to physical activity during the covid-19 pandemic’.

Introduction

Line 39 - “...in almost all countries worldwide”.

Line 48 – were there any allowances for individuals to have an hour or something similar of PA a day, or was it a complete lockdown and not allowed out at all?

Line 57 - levels.

Line 59 – state what SARS-CoV-2 stands for the first time you write it.

Line 61 – you mention PA promoting policies to be taken, this needs to be expanded on and the sentence flow better in terms of what you are trying to say.

Line 63 – related.

Line 73- “… physical, social and environmental factors…”.

Line 117 – levels.

Materials and Methods

Line 122 – add ‘years’ in the brackets as the unit.

Line 124 – what does ISCED stand for?

Line 158 – those who did not meet the guidelines, were they classed as inactive?

Lines 159 to 161 – negatives of self-reporting??

Combine the last two paragraphs within the methods section as both talk about the statistical analysis.

Results

Line 185 – PA write what it stands for, AF what is this?? Think it should be PA?? Always spell out in full within the key.

Lines 202 to 206 – please check these are accurate as you stated ISCED2 have more barriers related to ‘Lack of time’ however they were 42.8% compared to 55.4% (ISCED 3)?? ‘Takes away the time needed to study’ also seems inaccurate? ‘Lack of motivation/interest’??

Line 209 – title needs to go on same page as the table.

Line 219 – AF??

Table 6/Lines 223 to 226 – ‘I’m not in shape’ ‘Lack of ability’, ‘I’m ashamed’ ‘Too expensive’ ‘I don’t like competition’ ‘Lack of motivation/interest’. You have said the normal weight group selected showed significantly more PA barriers – but you have reported them here as showing less??

Discussion

Line 248 – were not was.

Line 254 – pandemics.

Lines 248 to 264 – you are required to sort the tables in the results section to ensure that the significant differences you mention here match.

Lines 284-296 – again the results need amending to reflect what you are saying here.

Line 311- regardless of the barrier type.

Lines 310 to 324, in relation to this paragraph could it also be that the normal weight adolescents have had their routine affected so they feel more barriers to PA, as their structure has been taken from them?

Lines 351-354 – work on the flow in the middle of this sentence.

Line “…PA barriers and strategies that should be …”??

Line 361 – levels.

Line 363 – data has a capital but no full stop before it??

Author Response

Response to Reviewer 1 Comments

Thank you so much for your comments and suggestions. The changes you propose are an important contribution to improving the quality of the article. Thank you!

Point 1: The title needs to read: ‘Adolescents’ perceived barriers to physical activity during the covid-19 pandemic’.

Response 1: I agree. Title has been changed.

Point 2: Introduction;

Line 39 - “...in almost all countries worldwide”

Response 2: I agree. Correction made.

Point 3: Line 48 – were there any allowances for individuals to have an hour or something similar of PA a day, or was it a complete lockdown and not allowed out at all?

Response 3: I agree, text was added to clarify.

Point 4: Line 57 - levels.

Response 4: I agree. Correction made.

Point 5: Line 59 – state what SARS-CoV-2 stands for the first time you write it.

Response 5: I agree, text was added to clarify.

Point 6: Line 61 – you mention PA promoting policies to be taken, this needs to be expanded on and the sentence flow better in terms of what you are trying to say.

Response 6: I agree, text has been added. I think it completes the sentence and the meaning that I wanted to put into it

Point 7: Line 63 – related.

Response 7: I agree. Correction made.

Point 8: Line 73- “… physical, social and environmental factors…”.

Response 8: I agree. Correction made.

Point 9: Line 117 – levels.

Response 9: I agree. Correction made.

Point 10: Materials and Methods

Line 122 – add ‘years’ in the brackets as the unit.

Response 10: I agree. Correction made.

Point 11: Line 124 – what does ISCED stand for?

Response 11: I agree, text was added to clarify.

Point 12: Line 158 – those who did not meet the guidelines, were they classed as inactive?

Response 12: Text was added to clarify. Only students who met the WHO guidelines were considered "active."

Point 13: Lines 159 to 161 – negatives of self-reporting??

Response 13: I am not sure if you are referring to the disadvantages of self-reporting. If so, that limitation is reported in the Discussion.

Point 14: Combine the last two paragraphs within the methods section as both talk about the statistical analysis.

Response 14: I agree. Correction made.

Point 15: Results

Line 185 – PA write what it stands for, AF what is this?? Think it should be PA?? Always spell out in full within the key.

Response 15: Correction made. It was a translation error.

Point 16: Lines 202 to 206 – please check these are accurate as you stated ISCED2 have more barriers related to ‘Lack of time’ however they were 42.8% compared to 55.4% (ISCED 3)?? ‘Takes away the time needed to study’ also seems inaccurate? ‘Lack of motivation/interest’??

Response 16: I understand the doubt. But the data are correct. The value in the brackets is the percentage within ISCED 2. That is, 42.8% of those belonging to ISCED 2. And the same with the other barriers you pointed out. I added a note for clarification (at the bottom of the table).

Point 17: Line 209 – title needs to go on same page as the table.

Response 17: I agree. Correction made.

Point 18: Line 219 – AF??

Response 18: Correction made. It was a translation error.

Point 19: Table 6/Lines 223 to 226 – ‘I’m not in shape’ ‘Lack of ability’, ‘I’m ashamed’ ‘Too expensive’ ‘I don’t like competition’ ‘Lack of motivation/interest’. You have said the normal weight group selected showed significantly more PA barriers – but you have reported them here as showing less??

Response 19: This topic is related to the clarification I made in point 16. I added a note to clarify at the bottom of the table.

Point 20: Discussion

Line 248 – were not was.

Response 20: I agree. Correction made.

Point 21: Line 254 – pandemics.

Response 21: I agree. Correction made.

Point 22: Lines 248 to 264 – you are required to sort the tables in the results section to ensure that the significant differences you mention here match.

Response 21: This topic is related to the clarification I made in point 16. I added a note to clarify at the bottom of the table.

Point 23: Lines 284-296 – again the results need amending to reflect what you are saying here.

Response 23: This topic is related to the clarification I made in point 16. I added a note to clarify at the bottom of the table.

Point 24: Line 311- regardless of the barrier type.

Response 24: I agree. Correction made.

Point 25: Lines 310 to 324, in relation to this paragraph could it also be that the normal weight adolescents have had their routine affected so they feel more barriers to PA, as their structure has been taken from them?

Response 25: I agree. Text has been added to adress this point.

Point 26: Lines 351-354 – work on the flow in the middle of this sentence.

Response 26: I agree. the text and punctuation were changed to make it clearer and more fluid.

Point 27: Line “…PA barriers and strategies that should be …”??

Response 27: I agree. Correction made.

Point 28: Line 361 – levels.

Response 28: I agree. Correction made.

Point 29: Line 363 – data has a capital but no full stop before it??

Response 29: I agree. Correction made.

Reviewer 2 Report

This manuscript provides important insights into the impact of the pandemic on adolescent's participation in physical activity.  The author(s) have done a thorough review of the literature and presented a good analysis of their findings.  Additional consideration would be useful of the impact of their data collection methods on potential findings.  Surveys were conducted of students in PE classes.  There is evidence to indicate that girls are less likely to participate in PE and therefore may have skewed the findings to illustrate further lower participation.  It would be useful to explore this in the limitations of the study and conclusion.  Also, it would be useful to explore further whether Portugal had any unique contextual factors as opposed to other countries during the pandemic and what might have been other similar experiences.  Overall, an interesting manuscript with important implications. 

Author Response

Response to Reviewer 2 Comments

Your comments were very accurate and definitely helped improve the quality of the article.

Thank you!

Point 1: Additional consideration would be useful of the impact of their data collection methods on potential findings. 

Response 1: Alguns tópicos foram adicionados na secção Discussão, no sentido de lançar linhas de investigação futuras que complementem os dados aqui encontrados.

Point 2: Surveys were conducted of students in PE classes.  There is evidence to indicate that girls are less likely to participate in PE and therefore may have skewed the findings to illustrate further lower participation.  It would be useful to explore this in the limitations of the study and conclusion. 

Response 2: I agree. Text added in agreement with the important question you raise.

Point 3: Also, it would be useful to explore further whether Portugal had any unique contextual factors as opposed to other countries during the pandemic and what might have been other similar experiences. 

Response 3: I agree. Different phases of the pandemic correspond to distinct measures to fight the virus. Text added text regarding this point.
